# Unveiling LoRA Intrinsic Ranks via Salience Analysis

Wenjun Ke[1,2], Jiahao Wang[1], Peng Wang[1,2*], Jiajun Liu[1], Dong Nie[3], Guozheng Li[1], and Yining Li[1]

[1]School of Computer Science and Engineering, Southeast University
[2]Key Laboratory of New Generation Artificial Intelligence Technology and Its Interdisciplinary Applications (Southeast University), Ministry of Education
[3]Meta Inc.
{kewenjun, wang_jh, pwang, jiajliu, gzli, liyining}@seu.edu.cn, dongnie@cs.unc.edu

## Abstract

The immense parameter scale of large language models underscores the necessity for parameter-efficient fine-tuning methods. Methods based on Low-Rank Adaptation (LoRA) assume the low-rank characteristics of the incremental matrix and optimize the matrix obtained from low-rank decomposition. Although effective, these methods are constrained by a fixed and unalterable rank, neglecting the variable importance of matrices. Consequently, methods for adaptive rank allocation are proposed, among which AdaLoRA demonstrates excellent fine-tuning performance. AdaLoRA conducts adaptation based on singular value decomposition (SVD), dynamically allocating ranks according to importance. However, it still struggles to achieve a balance between fine-tuning effectiveness and efficiency, leading to limited rank allocation space. Additionally, the importance measurement focuses only on parameters with minimal impact on the loss, neglecting the dominant role of singular values in SVD-based matrices and the fluctuations during training. To address these issues, we propose SalientLoRA, which unveils the intrinsic ranks of the weight matrix via salience measurement and adaptively optimizes ranks of LoRA. This method measures the salience of rank within a time-series by constructing inter-dependencies among the correlations of singular values and prune ranks with low salience while retaining those with high significance. Additionally, an adaptive adjustment of the time-series window enhances the speed of rank allocation while ensuring training stability. This mechanism enables matrics to set a higher initial rank, thus expanding the allocation space for ranks. To evaluate the generality of our method across various tasks, we conduct experiments on natural language understanding (NLU), natural language generation (NLG), and large model instruction tuning tasks. Experimental results demonstrate the superiority of SalientLoRA, which outperforms state-of-the-art methods by 0.96%-3.56% on multiple datasets. Furthermore, as the rank allocation space expands, our method ensures fine-tuning efficiency, achieving a speed improvement of 94.5% compared to AdaLoRA. The code is publicly available at https://github.com/Heyest/SalientLoRA.

## 1 Introduction

Large language models (LLMs) [23, 28, 2, 4] exhibit robust generative and inferential capabilities, excelling in various downstream tasks [26, 12, 14, 16]. However, the vast number of parameters in LLMs makes fine-tuning computationally demanding and time-consuming. Notably, LLaMA [23] encompasses parameters ranging from 7 billion to 65 billion, and fine-tuning LLaMA 65B necessitates a substantial 780GB of GPU memory [5].

---

*Corresponding author.

38th Conference on Neural Information Processing Systems (NeurIPS 2024).

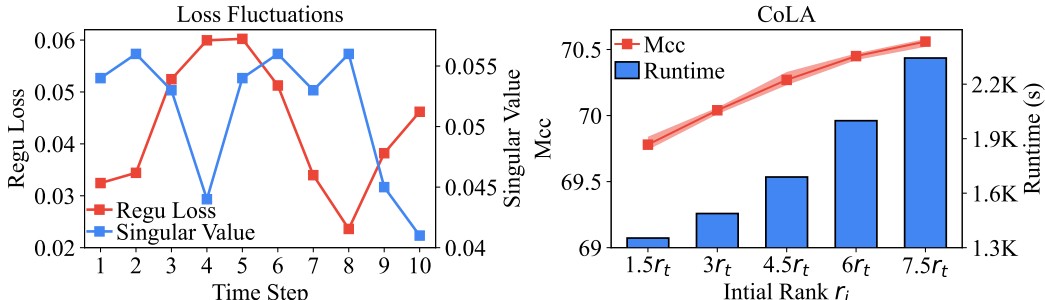

Figure 1: (Left) Fluctuations of regularization loss and singular values across multiple time steps. (Right) Performance and runtime of fine-tuning DeBERTaV3-base model on the CoLA dataset with increased initial rank in AdaLoRA. Here, the metric Mcc means Matthews Correlation Coefficient.

To improve fine-tuning efficiency, numerous parameter-efficient fine-tuning (PEFT) methods [9, 13, 18, 10, 29, 17, 34] have been developed with only a minimal number of trainable parameters, significantly reducing computational resources. Incremental methods [9, 13, 18] introduce extra trainable parameters into the existing architecture of LLMs. However, the increased model depth introduces time delays during inference. Reparameterization methods [10, 29, 17, 34, 15], based on low-rank adaptation (LoRA) [10], assume the low-rank nature of the incremental matrix and perform low-rank decomposition on it. Although effective, they are still constrained by a fixed intrinsic rank, potentially limiting their adaptability to the dynamic significance of different matrix elements.

To address the aforementioned limitations, dynamic rank allocation methods [25, 6, 29] have been proposed, which can be seamlessly incorporated into LoRA-based PEFT approaches to significantly enhance adaptability. Among these methods, AdaLoRA [29] stands out, having demonstrated excellent fine-tuning performance, as evidenced by its widespread adoption in numerous research studies [33]. AdaLoRA [29] introduces singular value decomposition (SVD) to the incremental matrices, employing sophisticated importance-aware methods [30, 31] to adaptively allocate intrinsic ranks. This method improves the alignment of ranks with the dynamic importance of matrix elements. However, AdaLoRA still exhibits limitations in assessing the importance of LoRA ranks and fine-tuning efficiency in the following two aspects: (1) The importance assessment focuses only on parameters with minimal impact on loss, thereby neglecting the dominant role of singular values in the SVD matrix. This assessment also fails to consider the variability of parameters across multiple time steps, rendering it vulnerable to training fluctuations and probabilistic instability. Figure 1 (left) illustrates significant fluctuations in singular values and the regularization loss of orthogonality, with a sharp increase and decrease of regularization loss from steps 3 to 8. Such variability compromises the reliability of the importance measurement. (2) AdaLoRA starts with a specific initial rank and adjusts rank allocation by trimming unimportant parameters in groups during the fine-tuning process. Despite this, it struggles to balance fine-tuning efficiency and effectiveness, resulting in a limited rank allocation space. Figure 1 (right) illustrates that increasing the initial rank $r_i$ from 1.5 to 7.5 times the target rank $r_t$ improves fine-tuning performance by 0.78%. However, as the rank allocation space expands, the fine-tuning time nearly doubles.

In this paper, we argue that in LoRA optimization, the inter-dependencies among multidimensional ranks are crucial for unveiling the intrinsic ranks, especially when a specific rank significantly influences others. As a consequence, we propose SalientLoRA, a method that conducts salience analysis within a time-series LoRA ranks, pruning ranks with low salience while retaining those with high significance. Following AdaLoRA, we decompose the incremental matrix using SVD through regularization constraints. The salience measurement calculates the correlation between singular values across modules in a time-series and constructs a dependency graph of correlation relationships. This graph reveals the inter-dependencies of ranks, assigning higher salience to rank with a broader influences domain. The salience measurement also examines the variation of the regularization loss of orthogonality and singular values in the time-series, mitigating instability and randomness that may arise during training. Moreover, to balance fine-tuning performance and efficiency, we propose an adaptive adjustment of time-series windows. This mechanism dynamically controls the size of time-series for salience measurement and rank reduction, facilitating rapid rank allocation while preserving training stability. This mechanism allows matrices to set a higher initial rank, thereby expanding the rank allocation space with greater efficiency. To validate the generality of our method, we conduct experiments on the GLUE dataset for NLU tasks, the XSum and CNN

datasets for NLG tasks, and the MT dataset for instruction tuning, separately fine-tuning encoder-only (DeBERTaV3-base), encoder-decoder (BART-large and T5-base), and decoder-only (LLaMA) models. Experimental results demonstrate the superiority of our approach, which outperforms other existing fine-tuning methods by 0.96%-3.56% across multiple datasets, achieving state-of-the-art results. Moreover, as the rank allocation space expands, our method ensures fine-tuning efficiency, achieving a speed improvement of 94.5% compared to AdaLoRA.

## 2 Related Work

### 2.1 Parameter-Efficient Fine-Tuning

Parameter-efficient fine-tuning methods (PEFT) can be divided into two categories: incremental [9, 13, 18] and reparameterization approaches [10, 29, 17]. Incremental methods add small neural modules into the existing architecture of LLMs, focusing solely on training these newly integrated modules. Adapter-Tuning [9] inserts simple Adapter modules after the feedforward layer of each Transformer architecture. Prefix-Tuning [13] and Prompt-Tuning [18] introduce additional trainable prefix vectors before the input layer or hidden layers. However, these approaches introduce time delays during inference due to either increasing model depth or reducing the available input sequence length. Additionally, they still exhibit performance discrepancies compared to full-parameter fine-tuning approaches. Reparameterization techniques update the incremental weight matrix in a parameter-efficient manner, without altering the existing architecture of models. LoRA [10] assumes the low-rank nature of the incremental matrix and performs low-rank decomposition on it. DoRA [17] decomposes pre-trained weights into magnitude and direction for fine-tuning, using low-rank decomposition for directional updates. Although the LoRA-based reparameterization approaches are effective, they assign the same rank to all weight matrices, failing to consider the varying importance of weight matrices across different layers and modules.

### 2.2 Dynamic Rank Allocation

Intuitively, more important matrices are capable of learning more complex knowledge and thus require a higher rank. Therefore, the dynamic rank allocation methods [25, 6, 29] are necessary, which can be seamlessly applied to the LoRA-based approaches. To achieve the dynamic low-rank adaptation, DyLoRA [25] trains LoRA blocks for a range of ranks by sorting the representation learned by the adapter module at different ranks during training. SoRA [6] controls rank cardinality under gate sparsity by integrating a gate unit optimized through the proximal gradient. AdaLoRA [29] performs adaptation through singular value decomposition (SVD), dynamically adjusting intrinsic ranks based on significance. Among these methods, AdaLoRA has demonstrated its effectiveness, as proved by numerous studies and experiments. However, AdaLoRA still exhibits limitations in assessing the importance of LoRA ranks and fine-tuning efficiency.

## 3 Overview

As illustrated in Figure 2, the overall architecture of our method comprises three parts: incremental matrix SVD, salience measurement and adaptive rank allocation.

Firstly, given the pretrained weight $\mathbf{W}_0 \in \mathbb{R}^{d_1 \times d_2}$, the update process can be formalized as $\mathbf{W} = \mathbf{W}_0 + \boldsymbol{\Delta}$, where $\boldsymbol{\Delta}$ represents the incremental updates. To reduce the parameter count and facilitate rank allocation, we decompose $\boldsymbol{\Delta}$ into $\mathbf{PVQ}$, where $\mathbf{V} \in \mathbb{R}^{r \times r}$ is a diagonal matrix, $\mathbf{P} \in \mathbb{R}^{d_1 \times r}$ and $\mathbf{Q} \in \mathbb{R}^{r \times d_2}$, with rank $r \ll \{d_1, d_2\}$. Following AdaLoRA, the orthogonality of $\mathbf{P}$ and $\mathbf{Q}$ is regularized through the auxiliary loss to simulate singular value decomposition (SVD). The regularization loss $R(\mathbf{P}, \mathbf{Q})$ is defined as follows:

$$R(\mathbf{P}, \mathbf{Q}) = ||\mathbf{P}^T \mathbf{P} - \mathbf{I}||_{\mathrm{F}}^2 + ||\mathbf{Q}\mathbf{Q}^T - \mathbf{I}||_{\mathrm{F}}^2 \tag{1}$$

Through this decomposition approach, the rank of the incremental matrix can be easily controlled by zeroing out the singular values in $\mathbf{V}$. Secondly, we measure the salience of singular values within a time-series through orthogonality-aware singular value magnitudes and the influence domain, unveiling the inter-dependencies among ranks. Thirdly, singular values are sorted by their salience at the last step of each time-series, and those with low significance are zeroed out to achieve the

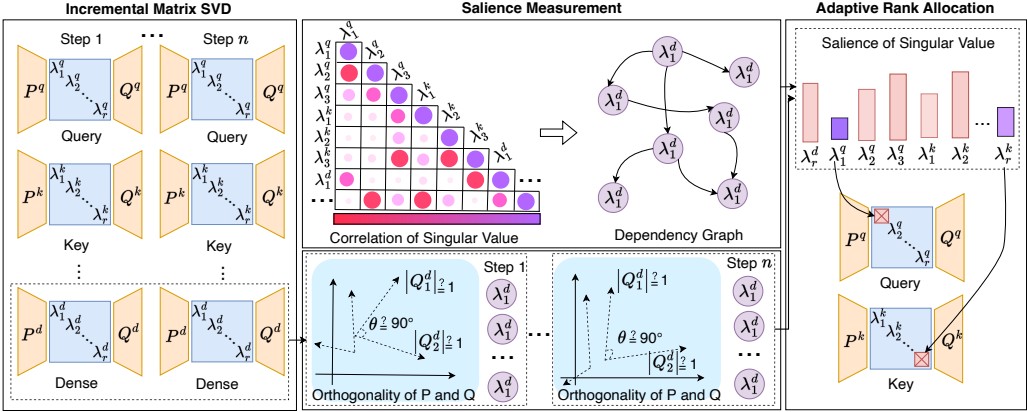

Figure 2: The overall framework of SalientLoRA. First, the incremental matrix is decomposed using SVD to facilitate rank allocation. During fine-tuning, the salience of singular values within time-series is measured, which is composed of orthogonality-aware singular value magnitudes and the influence domain of the dependency graph. Finally, in the rank allocation process, singular values with lower significance are progressively trimmed.

adaptive rank allocation. The rank allocation space is determined by the initial total rank $r_i$ and the target total rank $r_t$, both of which are hyperparameters. Upon completion of the rank allocation, the most critical $r_t$ ranks are retained. In this process, an adaptive time-series window is introduced to accelerate the rank allocation process and maintain stability during training. Section 4 and 5 elaborate on the Salience Measurement and the Adaptive Time-series Window, respectively.

## 4 Salience Measurement

### 4.1 Orthogonality-Aware Singular Value Magnitudes

Following the objective of SVD, which utilizes singular values to represent the primary characteristics of a matrix, we assess their importance based on the magnitudes. However, the reliability of SVD heavily relies on the orthogonality of $\mathbf{P}$ and $\mathbf{Q}$. Inadequate orthogonality undermines the efficacy of importance assessment with magnitudes. Therefore, we consider variations of regularization losses within a time-series, assigning weights to singular value magnitudes according to the orthogonality. Specifically, for the singular value $\lambda_a$, we record its magnitude $\lambda_a = \{\lambda_a^{(1)}, \lambda_a^{(2)}, \ldots, \lambda_a^{(n)}\}$ and regularization losses of the corresponding incremental matrix $R(\mathbf{P}_a^{(1)}, \mathbf{Q}_a^{(1)}), R(\mathbf{P}_a^{(2)}, \mathbf{Q}_a^{(2)}), \ldots, R(\mathbf{P}_a^{(n)}, \mathbf{Q}_a^{(n)})$ within the $n$ time steps. A high regularization loss indicates low reliability of the SVD at that step, and consequently, the weight assigned to it should be small. Therefore, the weight of step $i$ is calculated as follows:

$$w^{(i)} = \frac{\sum_{j=0}^n R(P_a^{(j)}, Q_a^{(j)})}{R(P_a^{(i)}, Q_a^{(i)})} \tag{2}$$

Subsequently, we normalize the weights with Min-Max normalization, then calculate the orthogonality-aware singular values magnitudes:

$$m_a = \sum_{j=1}^n w^{(j)} \lambda_a^{(j)} \tag{3}$$

### 4.2 Influence Domain of Singular Value

During training, there exist dependency relationships among singular values, wherein the variation of one singular value can influence multiple others. In light of this, we construct the dependency graph of the correlations between singular values and determine their significance through the influence domain. Initially, the correlation is calculated between any pair of singular values within the $n$ time

windows $\lambda_a = \{\lambda_a^{(1)}, \lambda_a^{(2)}, \ldots, \lambda_a^{(n)}\}$ and $\lambda_b = \{\lambda_b^{(1)}, \lambda_b^{(2)}, \ldots, \lambda_b^{(n)}\}$ by the Pearson coefficient:

$$p_{ab} = \frac{Cov(\lambda_a, \lambda_b)}{\sqrt{D(\lambda_a, \lambda_b)}} \tag{4}$$

If the $p_{ab}$ exceeds $\beta$, it indicates a correlation between $\lambda_a$ and $\lambda_b$. For a correlated pair of singular values, the slope $k_{ab}$ between them is further computed through linear regression fitting to determine the extent of influence of $\lambda_a$ on $\lambda_b$. A larger slope $k_{ab}$ indicates a greater influence of $\lambda_a$ on $\lambda_b$, signifying a dependency relationship between them. Thus, we obtain the dependency matrix $D \in \mathbb{R}^{m \times m}$ and the element $D_{ab}$ of $D$ is calculated as follows:

$$D_{ab} = \begin{cases} k_{ab} & p_{ab} > \beta \text{ and } k_{ab} > \gamma \\ 0 & \text{else} \end{cases} \tag{5}$$

where $m$ denotes the total number of singular values and $\gamma$ is the threshold for filtering dependency relationships. Note that the Pearson correlation coefficients and slopes are calculated using matrix parallel multiplication, incurring only minimal computational and time costs. Subsequently, we transform matrix $D$ into a dependency graph, with singular values as nodes and dependency relationships as edges. The dependency graph among singular values forms a directed cyclic graph, containing numerous redundant dependencies within the cycles. Therefore, we employ a depth-first search (DFS) algorithm for de-cycling to eliminate these redundancies. The pseudocode of de-cycling algorithm is provided in Algorithm 1.

Finally, based on the dependency graph, the influence domain of singular values is computed:

$$I_i = \begin{cases} 1 & \text{node } i \text{ without subsequent nodes} \\ \sum_k D_{ik} I_k & \text{node } i \text{ with subsequent node } k \end{cases} \tag{6}$$

As illustrated in the dependency graph of Figure 2, some singular value determines multiple others, with its variation can consequently lead to changes in several other singular values, thus signifying a higher level of importance.

Ultimately, we combine the influence domain with orthogonality-aware magnitudes to obtain the salience for each singular value. Here, the $\lambda$ is a hyperparameter that controls the contribution of two components to the salience measurement. The salience of the $\lambda_a$ is denoted as $s_a$:

$$s_a = \lambda m_a + (1 - \lambda) I_a \tag{7}$$

---

**Algorithm 1** De-Cycling Algorithm for Dependency Graphs

---

**Input:** Dependency graph $G$
**Output:** Directed acyclic graph $G'$
 1: $Path = stack()$      ▷ Initialize a stack to record the traversal path
 2: $Visited = \varnothing$      ▷ Record the nodes that have been visited
 3: **for** each node $v$ in $G$ **do**
 4:   **if** $v$ not in $Visited$ **then**
 5:    DFS_VISIT($v$)
 6:   **end if**
 7: **end for**
 8:
 9: **def** DFS_VISIT($v$)
10:   $Visited$.add($v$)
11:   $Path$.push($v$)
12:   **for** each node $u$ adjacent to $v$ **do**
13:    **if** $u$ not in $Visited$      ▷ If $u$ is in the path, a cycle has been detected.
14:     DFS_VISIT($u$)
15:    **else if** $u$ in Path
16:     Remove the edge with the smallest weight in the cycle
17:    **end if**
18:   **end for**
19:   $Path$.pop(v)

---

# 5 Adaptive Time-Series Window

During the rank allocation for $n_f$ steps, we implement an adaptive adjustment of time-series window. Within each time-series, the salience of singular values is assessed and the unimportant singular values are dropped at the last step of the time-series. This mechanism follows the principle that during the early stages of training, rank adjustment can be swiftly executed, while in the later stages, rank allocation should be approached more cautiously. Specifically, we initially maintain a small time window during the early stages of training, facilitating rapid rank reduction to enhance rank allocation efficiency. The pruned singular values and their corresponding singular vectors are excluded from gradient updates and salience calculations to expedite fine-tuning. As training progresses, the time window expands, and the allocation becomes more cautious. The broad time window reduces the frequency of importance calculations, thereby further accelerating the fine-tuning process. The adjustment of the time window is as follows:

$$T = T_f + (T_i - T_f)(1 - \frac{n_t}{n_f})^3 \qquad (8)$$

where $T, T_i$, and $T_f$ represent the current time window size, the initial time window size, and the final time window size during the rank allocation phase, respectively, with $T_i < T_f$. $n_t$ denotes the current training step during the rank allocation phase.

The model starts with a relatively high initial rank $r_i$ and zeros out a certain number of singular values within each time window to allocate rank. At the end of the rank allocation process, the rank is pruned to the target total rank $r_t$. The total rank count maintained in each time window is:

$$r = r_i - \frac{T}{T_f} \times (r_i - r_t) \qquad (9)$$

# 6 Experiment

## 6.1 Experimental Settings

**Datasets**    To evaluate the applicability of our fine-tuning approach across multiple tasks and various models, we conduct experiments on natural language understanding (NLU), natural language generation (NLG), and large-scale model instruction fine-tuning tasks, respectively. The specific datasets chosen for each task and the statistics will be detailed in Section 6.2, 6.3, 6.4 and Appendix A.

**Baselines**    We choose full fine-tuning (Full FT) and several existing incremental and reparameterization methods as baselines. Full FT denotes fine-tuning all parameters of the model, requiring significant computational resources. Adapter [9] integrates additional neural network layers into the model, only fine-tuning the newly added parameters. AdapterFusion [20] propose a more efficient design with adapters only applied after FFN and LayerNorm modules [3]. LoRA [10] performs low-rank decomposition on the incremental parameter matrices, substantially reducing the number of parameters for fine-tuning. SoRA [6] introduces a gating module with a proximal gradient decent update to control the sparsity of the updated matrices. AdaLoRA [29] dynamically distributes the parameter budget among weight matrices by evaluating the importance. DoRA [17] decomposes the pre-trained weights into magnitude and direction for fine-tuning, using LoRA for directional updates.

**Settings**    Our experiments are conducted on four NVIDIA RTX 3090Ti GPUs for NLU and NVIDIA Ampere A100 for NLG and instruction tuning tasks. During salience measurement, the slope threshold for dependency calculation $\gamma = 2$, the correlation threshold $\beta = 0.9$, and the $\lambda$ is set to 0.7. For adaptive time-window adjustment, the initial time window size $T_i = 10$, the final time window size $T_f = 200$, and the initial total rank $r_i$ is set to 7.5 times the target total rank $r_t$. The selected weight matrix of transformer layer for fine-tuning includes query/key/value projection ($W_q, W_k, W_v$), output projection ($W_o$) in the self-attention, and two weight matrices ($W_{f_1}, W_{f_2}$) in two feedforward layers (FFNs). We select the learning rate from $\{8 \times 10^{-5}, 5 \times 10^{-5}, 3 \times 10^{-5}, 1 \times 10^{-4}, 3 \times 10^{-4}, 5 \times 10^{-4}, 8 \times 10^{-4}, 1 \times 10^{-3}\}$, and pick the best-performing learning rate for every method. Further details on other hyperparameters are shown in Appendix C.

## 6.2 Natural Language Understanding

For NLU, we evaluate our method on the GLUE [27] benchmark utilizing the encoder-only model DeBERTaV3-base [7]. The GLUE benchmark consists of eight datasets: CoLA, SST-2, MRPC, QQP, STS-B, MNLI, QNLI, and RTE. We use Matthew's correlation coefficient, Spearman's correlation coefficient, and overall accuracy to evaluate the CoLA, STS-B, and MNLI datasets. For the remaining datasets, we apply accuracy as the evaluation metric. To compare under the same parameter budget, we set the hidden dimensions $d$ of adapters to 8 and 32, the rank $r$ of LoRA and DoRA to 2 and 8, and the target total ranks $r_t$ for AdaLoRA and SalientLoRA to 144 and 276, respectively.

Table 1: Experimental results of SalientLoRA and other baselines on the GLUE benchmark across varying parameter budgets. The bold scores indicate the best results. We report the average performance over 5 runs using different random seeds, with SalientLoRA significantly better than AdaLoRA and DoRA with p-value < 0.05 based on paired t-test.

| Method | #Params | MNLI | SST-2 | CoLA | QQP | QNLI | RTE | MRPC | STS-B | Ave. |
|---|---|---|---|---|---|---|---|---|---|---|
| Full FT | 184M | 89.98 | 95.64 | 69.21 | 92.05 | 93.78 | 82.49 | 89.22 | 91.59 | 88.00 |
| Adapter$_{d=8}$ | 0.31M | 90.10 | 95.41 | 67.65 | 91.54 | 93.52 | 83.39 | 89.25 | 91.31 | 87.60 |
| AdapterFusion$_{d=8}$ | 0.30M | 89.89 | 94.72 | 69.06 | 91.40 | 93.87 | 84.48 | 89.71 | 91.38 | 87.90 |
| LoRA$_{r=2}$ | 0.32M | 90.04 | 94.95 | 68.71 | 91.61 | 94.01 | 85.31 | 89.65 | 91.58 | 88.23 |
| DoRA$_{r=2}$ | 0.34M | 90.14 | 95.78 | 70.21 | 91.77 | 94.17 | 87.48 | 90.27 | 91.24 | 88.88 |
| AdaLoRA$_{r_t=144}$ | 0.32M | 90.22 | 95.76 | 70.04 | 91.78 | 94.13 | 87.36 | 90.13 | 91.21 | 88.83 |
| SalientLoRA$_{r_t=144}$ | 0.32M | **90.94** | **96.23** | **71.87** | **91.97** | **94.83** | **88.43** | **91.68** | **92.36** | **89.79** |
| Adapter$_{d=32}$ | 1.21M | 90.13 | 95.53 | 68.64 | 91.91 | 94.11 | 84.48 | 89.95 | 91.48 | 88.28 |
| AdapterFusion$_{d=32}$ | 1.18M | 90.33 | 95.61 | 68.77 | 92.04 | 94.29 | 85.20 | 89.46 | 91.54 | 88.41 |
| SoRA | 0.91M | 90.35 | 95.64 | 71.48 | 92.39 | 94.28 | 87.77 | 91.98 | 92.22 | 89.36 |
| LoRA$_{r=8}$ | 1.27M | 90.47 | 95.67 | 69.73 | 91.95 | 93.76 | 85.32 | 89.71 | 91.86 | 88.56 |
| DoRA$_{r=8}$ | 1.29M | 90.37 | 96.02 | 71.46 | 92.36 | 94.47 | 87.74 | 91.15 | 92.12 | 89.46 |
| AdaLoRA$_{r_t=276}$ | 1.27M | 90.27 | 95.95 | 70.86 | 92.13 | 94.28 | 87.36 | 90.22 | 91.39 | 89.06 |
| SalientLoRA$_{r_t=276}$ | 1.27M | **91.07** | **96.58** | **72.68** | **92.41** | **95.04** | **88.93** | **92.34** | **92.76** | **90.23** |

The results are shown in Table 1. Firstly, compared to incremental methods, reparameterization approaches generally yield better results. Under the parameter budget of 0.3M, LoRA and DoRA outperform adapters by average scores of 0.63% and 1.28% respectively. This demonstrates the superiority of reparameterization methods, achieving enhanced fine-tuning performance without adding extra inference latency. Secondly, both AdaLoRA and SalientLoRA consistently outperform LoRA across all eight datasets, showing an average improvement of 0.6% and 1.56% in 0.32 million parameters, and 0.5% and 1.67% in 1.27 million parameters, respectively. This suggests that adaptively adjusting the rank based on the importance of matrices indeed leads to better fine-tuning performance. Third, SalientLoRA surpasses all other fine-tuning methods with margins of 0.91%-1.79% and 0.77%-1.95% under two parameter budgets, achieving state-of-the-art results. Specifically, SalientLoRA outperforms AdaLoRA by 0.96% and 1.17% on average, which indicates the superiority of our method for salience measurement.

## 6.3 Natural Language Generation

For the NLG task, we finetune the encoder-decoder model BART-large [11] and T5-base [21] on the text summarization datasets XSum [19] and CNN/DailyMail [8], with evaluation metrics of ROUGE 1/2/L scores. We set the rank $r$ of LoRA and DoRA to 2, and the target total ranks $r_t$ for AdaLoRA and SalientLoRA to 144, respectively. Fine-tuning results on two datasets are shown in Table 2. The experimental results indicate that under the same parameter budget, SalientLoRA achieves the best performance when fine-tuning T5-base and BART-large, surpassing other methods by margins of 0.36%-2.2%. Specifically, when fine-tuning the BART-large model on the XSum dataset, SalientLoRA exceeds AdaLoRA by 0.64%, 0.86%, and 1% in ROUGE 1/2/L scores, respectively, and outperforms DoRA by 0.54%, 0.36%, and 0.65%. Moreover, SalientLoRA even surpasses the performance of full fine-tuning on the CNN/DailyMail dataset, with improvements of 1.42%, 0.18%, and 1.51% by T5-base and 0.78%, 0.58%, and 0.88% by BART-large, respectively.

Table 2: Performance comparison of different fine-tuning methods on NLG tasks. The three metrics on each dataset are ROUGE 1/2/L scores.

| Model | Method | #Params | XSum | | | CNN/DailyMail | | |
|-------|--------|---------|------|------|------|------|------|------|
| T5-base | Full FT | 212.6M | 38.81 | 16.50 | 31.27 | 42.05 | 20.34 | 39.40 |
| | LoRA | 0.34M | 36.35 | 14.16 | 28.88 | 41.27 | 19.33 | 38.76 |
| | AdaLoRA | 0.34M | 36.68 | 14.27 | 28.13 | 41.53 | 19.52 | 39.01 |
| | SoRA | 0.46M | 36.87 | 14.54 | 28.32 | 41.78 | 19.86 | 40.18 |
| | DoRA | 0.36M | 36.89 | 14.68 | 28.49 | 42.92 | 20.03 | 40.28 |
| | SalientLoRA | 0.34M | **37.36** | **15.03** | **29.14** | **43.47** | **20.52** | **40.91** |
| BART-large | Full FT | 375.5M | 45.49 | 22.33 | 37.26 | 44.16 | 21.28 | 40.90 |
| | LoRA | 0.60M | 42.81 | 19.68 | 34.73 | 43.68 | 20.63 | 40.71 |
| | AdaLoRA | 0.60M | 43.29 | 19.95 | 35.04 | 43.94 | 20.83 | 40.96 |
| | SoRA | 0.72M | 43.46 | 20.27 | 35.28 | 44.21 | 21.21 | 41.18 |
| | DoRA | 0.64M | 43.39 | 20.45 | 35.39 | 44.35 | 21.34 | 41.34 |
| | SalientLoRA | 0.60M | **43.93** | **20.81** | **36.04** | **44.94** | **21.86** | **41.78** |

## 6.4 Instruction Tuning

To evaluate the effectiveness of our method in fine-tuning large language models of the decoder-only architecture, we fine-tune the LLaMA-7B [23] and LLaMA2-7B [24] model on the Alpaca [22] instruction dataset, which consists of 52k instances generated by GPT-4 [1] based on inputs from Alpaca. The fine-tuned LLaMA and LLaMA2 are evaluated on the MT-Bench [32], generating model responses to a pre-defined set of 80 high-quality, multi-turn questions. These responses are then assessed using GPT-4, which assigns a quantitative score on a scale of 10 to each answer. We set the rank $r$ of LoRA and DoRA to 64, and the target total ranks $r_t$ for AdaLoRA and SalientLoRA to 12280, respectively. We present the average scores alongside the number of trainable parameters in Table 3. All compared fine-tuning methods utilized only 159.9M to 163.7M trainable parameters, which is merely 2.4% of Full FT. Experimental results indicate that LoRA still lags behind Full FT by 0.23 and 0.2 points. Notably, SalientLoRA surpasses all fine-tuning methods, even outperforming Full FT by 0.12 and 0.17 points, demonstrating the superior performance of SalientLoRA.

Table 3: The average score on MT-Bench and trainable parameter count of LLaMA-7B after instruction tuning by different fine-tuning methods.

| Model | Method | #Params. | Score |
|-------|--------|----------|-------|
| LLaMA | Full FT | 6426.3M | 5.12 |
| | LoRA | 159.9M | 4.89 |
| | AdaLoRA | 159.9M | 5.06 |
| | SoRA | 163.7M | 5.11 |
| | DoRA | 161.2M | 5.17 |
| | SalientLoRA | 159.9M | **5.28** |
| LLaMA2 | Full FT | 6426.3M | 5.25 |
| | LoRA | 159.9M | 5.05 |
| | AdaLoRA | 159.9M | 5.17 |
| | SoRA | 163.7M | 5.24 |
| | DoRA | 161.2M | 5.31 |
| | SalientLoRA | 159.9M | **5.42** |

## 6.5 Ablation Results

To validate the effectiveness of the two components in our salience measurement, we conduct experiments on the full SalientLoRA and its variants. Here, OAM represents the orthogonality-aware magnitude and ID represents the influence domain. SalientLoRA w/o OAM indicates using the mean of magnitudes within a time-series without adjusting weights based on orthogonality. We follow the experimental setup for the NLU task, setting the target total rank $r_t$ of SalientLoRA to 144. As shown in Table 4, we observe that both components of salience measurement (OAM and ID) significantly contribute to the final results. SalientLoRA w/o OAM experiences a performance decrease ranging from 0.18% to 0.73% across the eight datasets, with an average performance drop of 0.48%. This indicates that adaptively controlling the weights of magnitudes based on the matrix orthogonality effectively alleviates the problem of unreliable salience measurement caused by the orthogonality fluctuation during training. Additionally, SalientLoRA w/o ID shows a performance decline ranging from 0.32% to 1.13% across the eight datasets, with an average decrease of 0.8%. This highlights the crucial role of ID in performance enhancement. ID constructs dependency between singular values and further effectively measures importance by calculating the influence domain.

Table 4: The results of ablation experiments. Here, ↓ represents the performance declines of variants.

| Method | MNLI | SST-2 | CoLA | QQP | QNLI | RTE | MRPC | STS-B | Ave. | ↓ |
|---|---|---|---|---|---|---|---|---|---|---|
| SalientLoRA | 90.94 | 96.23 | 71.87 | 91.97 | 94.83 | 88.43 | 91.68 | 92.36 | 89.79 | - |
| w/o OAM | 90.36 | 95.87 | 71.14 | 91.79 | 94.28 | 88.04 | 91.05 | 91.94 | 89.31 | 0.48 |
| w/o ID | 90.12 | 95.59 | 70.56 | 91.65 | 94.11 | 87.52 | 90.63 | 91.76 | 88.99 | 0.8 |

## 6.6 Analysis of Space Allocation and Time Consumption

We remark that as the rank allocation space expands, both SalientLoRA and AdaLoRA exhibit performance improvements, and SalientLoRA achieves a better balance between performance and fine-tuning time due to the adaptive time-series window and parallelized computation. Given the resource-intensive nature of the experiments, we choose the CoLA, RTE, STS-B, and MRPC datasets from NLU tasks for validation. We set the target total rank $r_t$ at 144 and gradually increase the average initial rank $\bar{r}_i$ for each matrix from 3 to 18. Here, $\bar{r}_i = \frac{r_i}{n}$, where $n$ represents the number of matrices fine-tuned, and $r_i$ denotes the initial total rank. The results in Figure 3 indicate that as the initial rank increases, AdaLoRA shows performance improvements by 0.39% - 1.44%, while SalientLoRA improves by 0.55% - 1.5%. Furthermore, the fine-tuning time of AdaLoRA significantly rises by 108% on average, whereas SalientLoRA experiences only marginal increases, ranging from 7% to 9%. This highlights the superiority of our method in balancing fine-tuning performance and efficiency by the adaptive time-series window and parallelized computation during salience measurement, resulting in reduced fine-tuning time.

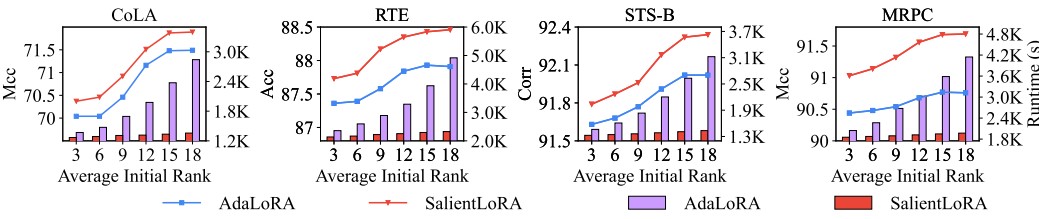

Figure 3: Comparison of fine-tuning effectiveness and runtime for AdaLoRA and SalientLoRA across multiple datasets as the rank space increases. The line graph illustrates their the fine-tuning performance, while the bar chart depicts the fine-tuning time.

## 6.7 Analysis of Hyperparameter $\lambda$

This section investigates the impact of the different contributions of two components in salience measurement on the final fine-tuning results, namely orthogonality-aware magnitude and influence domain. The degree of contribution is controlled by the hyperparameter $\lambda$. A higher $\lambda$ indicates a greater contribution of orthogonality-aware magnitude to the salience measurement. Conversely, as $\lambda$ decreases, the influence domain plays a greater role. The results in Figure 4 show an upward trend in fine-tuning performance across CoLA, RTE, STS-B, and MRPC datasets as $\lambda$ increases, reaching a peak when $\lambda$ reaches 0.7. This suggests that a greater contribution of orthogonality-aware magnitude leads to a more effective salience measurement. The primary reason lies in the SVD decomposition of the incremental matrix, where the magnitude of the singular values still represents the most fundamental factor of matrix characteristics. However, when $\lambda$ further rises, fine-tuning performance declines. This suggests that when the contribution of orthogonality-aware magnitude slightly outweighs that of the influence domain (i.e., when $\lambda$ approaches 0.7), salience measurement of singular values can achieve optimal performance, yielding the best fine-tuning results. This also demonstrates the effectiveness of both components in the salience measurement.

## 6.8 Salience Illustration across Different LoRA Ranks

In this section, we visualize the salience of singular values and analyze their distribution across different layers and modules in models. Here, we primarily illustrate the influence domain of singular values regarding inter-dependencies. Due to the presence of multiple singular values in each weight matrix, we average and normalize the influence domains for easier presentation. Figure 5 depicts the

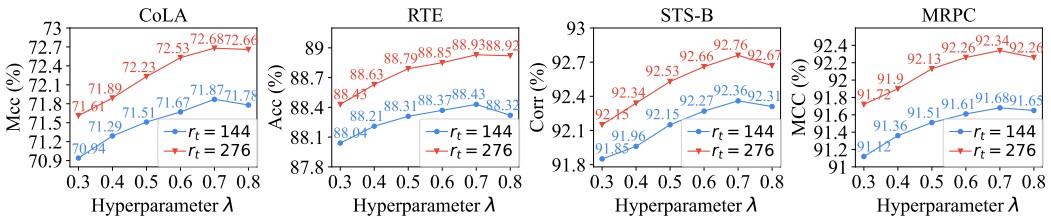

Figure 4: Variations of fine-tuning performance with different $\lambda$, which controls the contributions of two components in salience measurement: orthogonality-aware magnitude and influence domain.

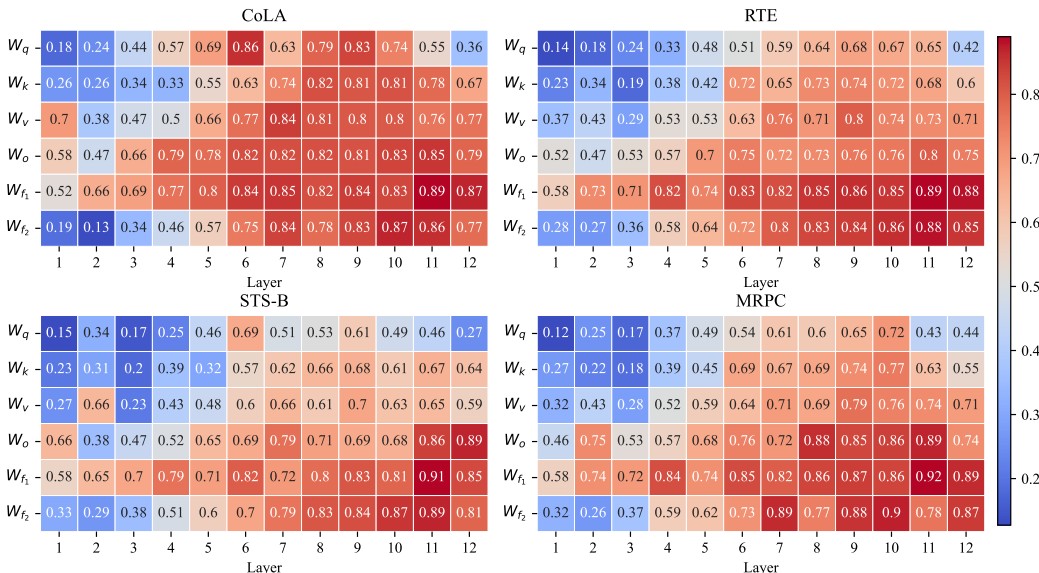

Figure 5: Distribution of influence domain across different layers and modules when fine-tuning DeBERTaV3-base model on different datasets.

distribution of the influence domains after the completion of rank allocation for DeBERTaV3-base on different datasets. The results reveal that deeper layers and Feedforward Network (FFN) modules generally possess larger influence domains compared to shallower layers and other components. This indicates that the variations of singular values in these modules can significantly influence other parameters, therefore possessing a higher importance. This observation also aligns with the empirical conclusions in AdaLoRA that weight matrices of FFN modules and deep layers are more crucial for model performance. Therefore, this validates the effectiveness of the influence domain for salience measurement to identify important parameters.

# 7   Conclusion

This paper proposes SalientLoRA, a novel adaptive rank allocation method for LoRA-based PEFT by salience analysis. This method measures the salience of rank within a time-series by constructing inter-dependencies among the correlations of singular values and prune ranks with low salience while retaining those with high significance, thereby unveiling the intrinsic ranks of the weight matrix. Additionally, an adaptive adjustment of the time-series window enhances the speed of rank allocation while ensuring training stability. Extensive experiments conducted in natural language understanding (NLU), natural language generation (NLG), and large model instruction tuning tasks demonstrate that our method achieves state-of-the-art fine-tuning performance, effectively balancing fine-tuning efficiency and performance.

## Acknowledgments

We thank the anonymous reviewers for their insightful comments. This work was supported by National Science Foundation of China (Grant Nos.62376057), the Start-up Research Fund of Southeast University (RF1028623234) and the Fundamental Research Funds for the Central Universities(2242024k30035). All opinions are of the authors and do not reflect the view of sponsors.

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

## A Dataset Statistics

The GLUE benchmark consists of eight datasets: CoLA, SST-2, MRPC, QQP, STS-B, MNLI, QNLI, and RTE. We use Matthew's correlation coefficient, Spearman's correlation coefficient, and overall accuracy to evaluate the CoLA, STS-B, and MNLI datasets. For the remaining datasets, we apply accuracy as the evaluation metric. Table 5 displays the statistics of the GLUE benchmark.

Table 5: The statistics of GLUE benchmark. Here, Mcc denotes Matthews correlation coefficient and Scc denotes Spearmans correlation coefficient.

| Dataset | #Train | #Test | #Dev | Metrics |
|---|---|---|---|---|
| Single-Sentence Classification | | | | |
| COLA | 8.5k | 1K | 1k | Mcc |
| SST | 67K | 872 | 1.8K | Accuracy |
| Pairwise Text Classification | | | | |
| MNLI | 393K | 20K | 20K | Accuracy |
| RTE | 2.5K | 276 | 3K | Accuracy |
| QQP | 364K | 40K | 391K | Accuracy |
| MRPC | 3.7K | 408 | 1.7K | Accuracy |
| QNLI | 108K | 5.7K | 5.7K | Accuracy |
| Text Similarity | | | | |
| STS-B | 7K | 1.5K | 1.4K | Scc |

The CNN/Daily Mail dataset serves as a corpus for single-document generative summarization, comprising news articles collected from CNN and Daily Mail, with each summary containing multiple summary sentences. XSum dataset consists of BBC articles and accompanying single sentence summaries. The statistics for the CNN/DailyMail and XSum datasets are shown in Table 6.

Table 6: The statistics of XSum and CNN/DailyMail datasets.

| Dataset | #Train | #Test | #Dev |
|---|---|---|---|
| CNN/DailyMail | 286k | 13K | 11k |
| XSum | 204K | 11k | 11K |

## B Experimental Analyses on Hyperparameter

We conduct additional experimental analyses on two sets of hyperparameters $\beta$, $\gamma$, $T_i$ and $T_f$ to explore their impact. These experiments are conducted on the CoLA and MRPC datasets to fine-tune the DeBERTaV3-base model. The total target rank is set to 144, and all the other parameters are consistent with the main experiments.

$\beta$ and $\gamma$ are thresholds that control the relevance and dependency relationships of singular values, respectively. To explore their effects, we keep other hyperparameters constant and change the values of $\beta$ and $\gamma$ separately to observe the timing and effects of SalientLoRA fine-tuning. The results are presented in the Table 7, where MCC represents Matthew's Correlation Coefficient and Acc denotes accuracy. The performance exhibits minimal sensitivity to variations in the values of $\beta$ and $\gamma$. This insensitivity stems from the decycling operation for the dependency graph , which effectively eliminates the majority of redundant dependency relationships. However, setting excessively low values for $\beta$ and $\gamma$ can lead to a large number of redundant dependencies in the graph, which increases the time cost of the decycling process and thus impacts the efficiency of fine-tuning. Therefore, we select hyperparameter values that ensure high efficiency of fine-tuning, with $\beta$=0.9 and $\gamma$=2.

Table 7: Experimental results with different values of hyperparameters and $\gamma$.

| $\beta$ | $\gamma$ | CoLA | | MRPC | |
|---|---|---|---|---|---|
| | | MCC | Time | Acc | Time |
| 0.5 | 2 | 71.87 | 29 | 91.65 | 42 |
| 0.7 | 2 | 71.82 | 24 | 91.57 | 34 |
| 0.9 | 2 | 71.87 | 21 | 91.68 | 33 |
| 0.9 | 2 | 71.78 | 27 | 91.63 | 39 |
| 0.9 | 1.5 | 71.83 | 34 | 91.67 | 47 |

$T_i$ and $T_f$ control the sizes of the initial and final time windows in the adaptive time-series window mechanism, respectively. The results in Table 8 indicate that the impact of $T_i$ on performance is minimal, with differences only ranging from 0.02% to 0.09%. However, as $T_i$ increases, the fine-tuning time significantly lengthens. This is due to a slower reduction in rank during the early stages of fine-tuning, which impacts the efficiency of rank allocation. Moreover, when $T_i$ remains constant and $T_f$ increases from 100 to 250, there is a slight improvement in performance, while the fine-tuning time remains relatively unchanged. This improvement can be attributed to the significance analysis in the later stages of fine-tuning, which incorporates singular values under more time steps, yielding more reliable allocation outcomes. Therefore, we set $T_i = 10$ and $T_f = 200$ to achieve a balance between performance and efficiency.

Table 8: Experimental results with different values of hyperparameters $T_i$ and $T_f$.

| $T_i$ | $T_f$ | CoLA | | MRPC | |
|---|---|---|---|---|---|
| | | MCC | Time | Acc | Time |
| 10 | 200 | 71.87 | 21 | 91.68 | 33 |
| 30 | 200 | 71.81 | 23 | 91.59 | 37 |
| 50 | 200 | 71.85 | 27 | 91.64 | 41 |
| 10 | 100 | 71.52 | 22 | 91.32 | 34 |
| 10 | 150 | 71.71 | 21 | 91.56 | 34 |
| 10 | 250 | 71.84 | 23 | 91.64 | 35 |

## C  Hyperparameter Statistics

The hyperparameter settings for NLU, NLG and instruction tuning tasks are shown in Table 9. Here, $r_t$ denotes the target total rank, $n_i$ represents the number of steps to warm up the training before rank allocation, and $n_f$ indicates the number of steps during the rank allocation phase.

Table 9: The hyperparameter settings for NLU, NLG and instruction tuning tasks.

| Task | Dataset | Learning Rate | Batch Size | Epochs | $r_t$ | $n_i$ | $n_f$ |
|---|---|---|---|---|---|---|---|
| NLU | MNLI | $5 \times 10^{-4}$ | 32 | 7 | 144/276 | 8000 | 35000 |
| | RTE | $8 \times 10^{-4}$ | 32 | 50 | 144/276 | 600 | 1500 |
| | QNLI | $8 \times 10^{-4}$ | 32 | 5 | 144/276 | 2000 | 8500 |
| | MRPC | $5 \times 10^{-4}$ | 32 | 30 | 144/276 | 600 | 1000 |
| | QQP | $3 \times 10^{-4}$ | 32 | 5 | 144/276 | 8000 | 25000 |
| | SST-2 | $8 \times 10^{-5}$ | 32 | 24 | 144/276 | 6000 | 20000 |
| | CoLA | $1 \times 10^{-3}$ | 32 | 25 | 144/276 | 800 | 2000 |
| | STS-B | $8 \times 10^{-4}$ | 32 | 25 | 144/276 | 800 | 2500 |
| NLG | XSum | $8 \times 10^{-4}$ | 64 | 25 | 144 | 6000 | 25000 |
| | CNN/DailyMail | $8 \times 10^{-4}$ | 32 | 15 | 144 | 5000 | 90000 |
| Instruction Tuning | Alpaca | $5 \times 10^{-4}$ | 4 | 1 | 12280 | 2000 | 6000 |

