# OpenReview forum: "Unveiling LoRA Intrinsic Ranks via Salience Analysis"
_NeurIPS.cc/2024/Conference — NeurIPS 2024 poster_

### Official Review · Reviewer_SXAJ · 2024-07-11

**Soundness:** 2
**Presentation:** 2
**Contribution:** 3
**Rating:** 5
**Confidence:** 4

**Summary:**

The work presents an algorithm for adapting the rank of the LORA matrices according to a novel “saliency metric”  assigned to each singular value of the LORA matrices.

The saliency measure is computed taking into account a sequence of steps (time window) during training and computing two quantities at the end of each time window: the orthogonality-aware singular values and the domain influence of each singular value. The orthogonality-aware singular value is a weighted average of the singular value where the weight takes into account the orthogonality of the SVD decomposition at that step. The domain influence takes into account the correlation between singular values within each time window.
At the end of each step sequence, the ranks are adjusted based on this salience measurement.

**Strengths:**

The authors propose a novel and interesting algorithm. The chosen setup speeds up the LoRA fine-tuning while maintaining accuracy or slightly outperforming other methods on the reported benchmarks. The experimental evaluation is convincing since the authors compare the proposed algorithm with other LoRA improvements on a reasonable number of tasks.

**Weaknesses:**

The main weakness of the work is the clarity of the exposition, which is obscure in some parts.

For example, one of the methods' core building blocks is the decycling operation of the dependency graph mentioned between lines 164-165: the authors must reference the algorithm they use for “de-cycling” the graph, describing its steps at least in the appendix.\
I leave other statements that require clarification in the question section below.

In addition, the paper does not discuss the limitations of the methods.

**Questions:**

> 1. *line 143 to 146: The authors claim that the weight assigned to singular values of high loss should be small. However, from equation (2), it seems that the steps with higher loss receive the larger weight in the time window.*

> 2.  *line 148. What do the authors mean by “we normalize the weights from 0 to 1”? An equation would be helpful to clarify the operation here.*

A key performance measure that is not discussed is memory consumption.
> 3. *Given a fixed parameter budget what is the amount of VRAM consumed by SalientLORA compared to for instance AdaLora?*

Other minor remarks:

When the authors state that AdaLORA has been adopted *in numerous research studies* (line 50) they should cite at least the most relevant ones to support the claim.

line 56 The sentence is somewhat obscure.
> 4. *What do the authors mean by “dominant role of singular values in the SVD matrix”? What is the precise meaning of “dominant role” in this context?*

**Limitations:**

No.
The authors do NOT seem to discuss the limitations of their method in the current version of the manuscript.

---

> ### Author Rebuttal · Authors · 2024-08-06
>
> We appreciate the constructive suggestions provided by the reviewer. We provide a detailed explanation and experimental analysis as follows.
>
> ***W1: the authors must reference the algorithm they use for “de-cycling” the graph, describing its steps at least in the appendix.***
>
> We provide a detailed description of the de-cycling process in the Global Response and present the algorithm's pseudocode in the PDF. This detailed description will be included in the final version of the paper.
>
> ***Q1: line 143 to 146: The authors claim that the weight assigned to singular values of high loss should be small. However, from equation (2), it seems that the steps with higher loss receive the larger weight in the time window.***
>
>  We sincerely apologize for the error of weight calculation formula presented in our paper. The correct formula  should be $w^{(i)}=\frac{\sum^n_{j=0}R(P^{(j)}_a,Q^{(j)}_a)}{R(P^{(i)}_a,Q^{(i)}_a)}$.  By calculating in this way, singular values with higher losses are assigned lower weights, thereby enhancing the effectiveness of importance assessment with magnitude in a time-series. We thank the reviewer for identifying this correction, and we will amend it in the final camera-ready version of the manuscript.
>
>
>
> ***Q2: Line 148. What do the authors mean by “we normalize the weights from 0 to 1”? An equation would be helpful to clarify the operation here.***
>
> We appreciate the reviewer’s suggestion. Here, we normalize the weights of the singular values to prevent large discrepancies in weight values from affecting the evalution. We use Min-Max normalization, and the calculation formula is as follows,
>
> $w^{(i)}_{norm}=\frac{w^{(i)}-w\_{min}}{w\_{max}-w\_{min}}$
>
>
> where $ w_{min}$ and $w_{max}$ represent the minimum and maximum values among $w^{(0)}$ to $w^{(n)}$, respectively. This normalization method ensures a uniform scale for all weights, thereby enhancing the robustness of our results. A comprehensive description of this normalization technique will be included in the camera-ready version of the manuscript.
>
>
>
> ***Q3: Given a fixed parameter budget what is the amount of VRAM consumed by SalientLORA compared to for instance AdaLora?***
>
> We appreciate the constructive suggestions provided by the reviewer. To compare the memory usage of AdaLoRA and SalientLoRA, we fine-tune the DeBERTaV3-base and BART-large models on the CoLA and XSum tasks respectively. We set the target rank $r_t$ to 144 and the initial total rank $r_i$ to 7.5 times $r_t$. The results in the table below indicate that the training overheads for both fine-tuning methods are quite close, with SalientLoRA exhibiting a marginal increase in memory occupancy of only 0.11%-0.22%. This slight difference arises because SalientLoRA needs to store all the singular values within the time window for salience assessment. However, as each matrix retains only 2-15 singular value, the impact on memory is minimal.
>
> |                       | CoLA    | XSum     |
> | --------------------- | ------- | -------- |
> | AdaLoRA $r_t=144$     | 5.084GB | 22.411GB |
> | SalientLoRA $r_t=144$ | 5.095GB | 22.436GB |
> | AdaLoRA $r_t=276$     | 5.399GB | 23.362GB |
> | SalientLoRA $r_t=276$ | 5.410GB | 23.387GB |
>
>
>
> ***Q4: When the authors state that AdaLORA has been adopted in numerous research studies (line 50) they should cite at least the most relevant ones to support the claim.***
>
> We are grateful for the reviewer's reminder. Numerous studies have shown that applying AdaLoRA achieves excellent fine-tuning performance in various scenarios, including speech models [3] and several other languages models [1, 2]. We will include these references in the final version of the paper.
>
>
> [1] Parameter-efficient fine-tuning methods for pretrained language models: A critical review and assessment. Xu L et al., 2023. In arXiv preprint.
>
> [2]  AutoPEFT: Automatic Configuration Search for Parameter-Efficient Fine-Tuning. Zhou et al., 2024. In Transactions of the Association for Computational Linguistics.
>
> [3] Whisper-med_15k. 2024. https://huggingface.co/sin2piusc/whisper-med_15k.
>
>
>
> ***Q5: What do the authors mean by “dominant role of singular values in the SVD matrix”? What is the precise meaning of “dominant role” in this context?***
>
> After performing singular value decomposition (SVD), the singular values represent the primary characteristics of a matrix. Larger singular values contain more information from the matrix and capture more of its essential features, thereby possessing greater importance. Consequently, singular values play a dominant role in the SVD matrix. In light of this, our method utilizes the magnitude of these singular values to assess their significance.

---

> > ### Comment · Reviewer_SXAJ · 2024-08-13
> >
> > I thank the authors for their response. I am inclined to keep my score.

---

> > > ### Author Response · Authors · 2024-08-13
> > >
> > > Dear Reviewer SXAJ,
> > >
> > > Thank you for your feedback.
> > >
> > > In your review comments, you indicated that the main drawback of the paper is the clarity of the presentation, with some details not being sufficiently explained. We have provided a detailed clarification for each of these aspects in our rebuttal, including the decycling process, normalization, and so on. These revisions will be incorporated into the final version of the paper to enhance its clarity. With these improvements, we believe the paper demonstrates better soundness and we hope you will consider raising your score. Your evaluation is very important to us, and we sincerely appreciate your consideration.

---

> > > > ### Comment · Reviewer_SXAJ · 2024-08-14
> > > >
> > > > I fully understand your situation and will consider your comment. However, addressing the reviewers' concerns ensures that the mark won't decrease, but it doesn't guarantee an increase. I still believe that this contribution is a borderline accept submission. I will discuss it further with the area chairs to help them make a decision.

---

> ### Author Response · Authors · 2024-08-08
> **We will include the following limitations in the camera-ready version.**
>
> In the final version of the paper, we will include the following limitations.
>
> The proposed mechanism of adaptive time-series window effectively accelerates and stabilizes the fine-tuning process. However, this mechanism employs a relatively rigid adaptive approach, considering only the time step information and focusing on progressively cautious pruning as the process advances. Future work will integrate the consideration of singular values' detailed dynamics, including momentum and interdependencies, to enable a more flexible and adaptive adjustment process.

---

> > ### Author Response · Authors · 2024-08-12
> >
> > Dear Reviewer SXAJ,
> >
> > Thank you for your thorough and valuable feedback. We have carefully addressed each of your concerns. We hope that our clarifications provide satisfactory answers to your questions. We are committed to improving our work based on your insights, and we look forward to your response.
> >
> > Best regards,
> > The Authors

---

> ### Author Response · Authors · 2024-08-14
>
> Thank you very much for your feedback. We appreciate your thorough and responsible review.
>
> Best regards.

---

### Official Review · Reviewer_i7vj · 2024-07-13

**Soundness:** 3
**Presentation:** 2
**Contribution:** 3
**Rating:** 5
**Confidence:** 3

**Summary:**

The paper introduces SalientLoRA, an approach designed to optimize the intrinsic ranks of LoRA components in LLMs through salience measurement. The method first utilizes salience measurement to analyze the variations and inter-dependencies of singular value magnitudes over time, which helps assess matrix importance while mitigating instability and randomness. This analysis informs the adaptive adjustment of the time-series window used for significance measurement and rank reduction during training. This adaptive mechanism allows for rapid and stable rank allocation, permitting an initially higher rank setting to expand the allocation space for ranks.

**Strengths:**

1. SalientLoRA's use of salience measurement to analyze and utilize the variations of singular values effectively addresses the challenges of instability and randomness in rank optimization. The adaptive adjustment of the time-series window for significance measurement during training enhances the efficiency and stability of rank allocation.

2.  Demonstrating substantial performance gains over state-of-the-art methods on diverse NLU and NLG tasks highlights the effectiveness of SalientLoRA in practical applications.

**Weaknesses:**

The proposed method incorporates a sophisticated multi-stage process that involves several critical hyperparameters, such as $\beta$, $\gamma$, $T_i$, and  $T_f$. However, the paper currently lacks a detailed analysis of these hyperparameters, which is crucial for understanding their roles and optimal settings within the methodology. Systematically exploring how each hyperparameter impacts the model's performance, including sensitivity analyses or hyperparameter tuning results, would greatly enhance the paper's scientific rigor.

**Questions:**

To fully evaluate the robustness of the proposed method, could you provide detailed ablation studies and analyses for the hyperparameters, including $\beta$, $\gamma$, $T_i$, and  $T_f$?

**Limitations:**

There are no potential negative societal impacts.

---

> ### Author Rebuttal · Authors · 2024-08-06
>
> ***Question: To fully evaluate the robustness of the proposed method, could you provide detailed ablation studies and analyses for the hyperparameters, including β, γ, Ti, and Tf?***
>
> We appreciate the insightful suggestions provided by the reviewer. In response, we conduct additional experimental analyses on two sets of hyperparameters to explore their impact. These experiments are conducted on the CoLA and MRPC datasets to fine-tune the DeBERTaV3-base model. The total target rank is set to 144, and all the other parameters are consistent with the main experiments.
>
> (1) $\beta$ and $\gamma$
>
> $\beta$ and $\gamma$ are thresholds that control the relevance and dependency relationships of singular values, respectively. To explore their effects, we keep other hyperparameters constant and change the values of $\beta$ and $\gamma$ separately to observe the timing and effects of SalientLoRA fine-tuning. The results are presented in the table below, where MCC represents Matthew’s Correlation Coefficient and Acc denotes accuracy.
>
> | $\beta$ | $\gamma$ | MCC (CoLA ) | Acc (MRPC) | Time (CoLA ) | Time (MRPC ) |
> | ------- | -------- | ----------- | ---------- | ------------ | ------------ |
> | 0.5     | 2        | 71.87       | 91.65      | 29min        | 42min        |
> | 0.7     | 2        | 71.82       | 91.57      | 24min        | 34min        |
> | 0.9     | 2        | 71.87       | 91.68      | 21min        | 33min        |
> | 0.9     | 1.5      | 71.78       | 91.63      | 27min        | 39min        |
> | 0.9     | 1        | 71.83       | 91.67      | 34min        | 47min        |
>
>
> The performance exhibits minimal sensitivity to variations in the values of $\beta$ and $\gamma$. This insensitivity stems from the decycling operation for the dependency graph , which effectively eliminates the majority of redundant dependency relationships. However, setting excessively low values for $\beta$ and $\gamma$​ can lead to a large number of redundant dependencies in the graph, which increases the time cost of the decycling process and thus impacts the efficiency of fine-tuning. Therefore, we select hyperparameter values that ensure high efficiency of fine-tuning, with $\beta$=0.9 and $\gamma$=2.
>
>
>
> (2) $T_i$ and $T_f$
>
> $T_i$ and $T_f$ control the sizes of the initial and final time windows in the adaptive time-series window mechanism, respectively. The results indicate that the impact of $T_i$ on performance is minimal, with differences only ranging from 0.02% to 0.09%. However, as $T_i$ increases, the fine-tuning time significantly lengthens. This is due to a slower reduction in rank during the early stages of fine-tuning, which impacts the efficiency of rank allocation.
>
> Moreover, when $T_i$ remains constant and $T_f$ increases from 100 to 250, there is a slight improvement in performance, while the fine-tuning time remains relatively unchanged. This improvement can be attributed to the significance analysis in the later stages of fine-tuning, which incorporates singular values under more time steps, yielding more reliable allocation outcomes. Therefore, we set $T_i=10$ and $T_f=200$ to achieve a balance between performance and efficiency.
>
> | $T_i$ | $T_f$ | MCC (CoLA ) | Acc (MRPC) | Time (CoLA ) | Time (MRPC ) |
> | ----- | ----- | ----------- | ---------- | ------------ | ------------ |
> | 10    | 200   | 71.87       | 91.68      | 21min        | 33min        |
> | 30    | 200   | 71.81       | 91.59      | 23min        | 37min        |
> | 50    | 200   | 71.85       | 91.64      | 27min        | 41min        |
> | 10    | 100   | 71.52       | 91.32      | 22min        | 34min        |
> | 10    | 150   | 71.71       | 91.56      | 21min        | 34min        |
> | 10    | 250   | 71.84       | 91.64      | 23min        | 35min        |
>
> #

---

> > ### Comment · Reviewer_i7vj · 2024-08-11
> >
> > Thank you for your responses! The responses address most of my concerns related to the hyperparameters. Hence, I will keep my original score.

---

> > > ### Author Response · Authors · 2024-08-12
> > >
> > > We are glad that our responses have addressed most of your concerns. We will incorporate this section into the final version of the paper to further enhance its rigor. If you find that the detailed hyperparameter analysis has contributed to improving the paper's soundness, we would be deeply grateful if you could consider raising the score of Soundness or the Overall Rating. Your evaluation is very important to us, and we sincerely appreciate your consideration.

---

> ### Author Response · Authors · 2024-08-11
>
> Dear Reviewer i7vj,
>
> Thank you for your thorough and valuable feedback. We have carefully addressed each of your concerns. We hope that our clarifications provide satisfactory answers to your questions. We are committed to improving our work based on your insights, and we look forward to your response.
>
> Best regards,
> The Authors

---

### Official Review · Reviewer_xsy4 · 2024-07-13

**Soundness:** 3
**Presentation:** 3
**Contribution:** 3
**Rating:** 6
**Confidence:** 4

**Summary:**

This paper proposes SalientLoRA, a new method for adaptively optimizing the intrinsic ranks of low-rank adaptation (LoRA) matrices. The key ideas are:

Using singular value decomposition (SVD) to decompose the LoRA matrices and measure the salience/importance of each singular value based on its magnitude, orthogonality constraints, and influence on other singular values within a time window during training.

**Strengths:**

- Novel salience measurement technique that considers singular inter-dependencies and temporal variations.
- Comprehensive evaluation across many datasets and model types (encoder, decoder, encoder-decoder).
- Achieves new state-of-the-art results on multiple benchmarks while being more efficient than prior LoRA methods.

**Weaknesses:**

- The article contains some details that are not clearly explained, such as how the R function on line 145 is calculated, and what specifically is done in the de-cycling process introduced on line 165.
- More analysis could be provided to interpret why the salience measurement works well. For example, are the average of influence domains consistent across models fine-tuned on different types of datasets?

**Questions:**

1. Taking the last row of Table 1 as an example, initially, the model uses a total rank that is 7.5 times the target rank, so the gpu memory usage is roughly equivalent to that of LoRA with r=8*7.5=60. Although the memory usage may decrease during the model optimization process, can you consider comparing with methods like LoRA and DoRA with r=60?
2. Based on my understanding, the constructed influence domains form an undirected simple graph. If this graph forms a single cycle, how do you perform de-cycling?
3. Do you calculate influence domains starting from vertices with a degree of 0, similar to topological sorting, and then update the degrees of the vertices connected to it, then repeat the process?

**Limitations:**

Yes

---

> ### Author Rebuttal · Authors · 2024-08-07
>
> We appreciate the constructive feedback provided by the reviewer. We provide a detailed explanation and experimental analysis as follows.
>
> ***W1: The article contains some details that are not clearly explained, such as how the R function on line 145 is calculated, and what specifically is done in the de-cycling process introduced on line 165.***
>
> (1) Function $R(\cdot)$
>
> In our paper, we have elaborated on the function $R(\cdot)$ for calculating the orthogonalization loss in Equation 1. Specifically,  $R(\cdot)$ measures the degree of orthogonality of $\textbf{P}$ and $\textbf{Q}$. The equation is as follows:
>
> $R(\textbf{P},\textbf{Q})=||\textbf{P}^T\textbf{P}-\textbf{I}||^2_F+||\textbf{Q}\textbf{Q}^T-\textbf{I}||^2_F$
>
> where $\textbf{P},\textbf{Q}$ denote the left and right singular matrices, respectively.
>
> (2) The De-cycling Process
>
> Due to space constraints, detailed content and analysis of the de-cycling process are provided in the Global Response.
>
>
> ***W2：More analysis could be provided to interpret why the salience measurement works well. For example, are the average of influence domains consistent across models fine-tuned on different types of datasets?***
>
> Due to space constraints, detailed content and analysis are provided in the Global Response.
>
> ***Q1: Taking the last row of Table 1 as an example, initially, the model uses a total rank that is 7.5 times the target rank, so the gpu memory usage is roughly equivalent to that of LoRA with r=8\*7.5=60. Although the memory usage may decrease during the model optimization process, can you consider comparing with methods like LoRA and DoRA with r=60?***
>
> We set the average target rank of each matrix in SalientLoRA to 8, and the initial rank to 60, to compare memory usage and fine-tuning performance with $\text{LoRA}\_{r=8}, \text{DoRA}\_{r=8}, \text{LoRA}\_{r=60}, \text{DoRA}_{r=60}$. Specifically, we fine-tune the DeBERTaV3-base and BART-large models on the CoLA and XSum datasets using the aforementioned settings.
>
> The results, as presented in the table below, demonstrate that small changes in rank settings do not significantly impact memory usage.  Memory usage of SalientLoRA peaks at only 5.41GB and 23.22GB, which are merely 0.33GB and 0.86GB higher than that of $\text{LoRA}\_{r=8}$.  Furthermore, the memory consumption of SalientLoRA rapidly decreases due to rank pruning, ultimately aligning closely with that of $\text{LoRA}\_{r=8}$. Notably, under this setting, our method still achieves the best results, surpassing those of $\text{LoRA}\_{r=60}$ and $ \text{DoRA}\_{r=60}$ by 0.9% and 0.57%, respectively. This also underscores the superior performance of our approach.
>
> | model       | GPU Mem (CoLA) | GPU Mem (XSum)  | Mcc (COLA) | Rough-L (XSum) |
> | ----------- | -------------- | --------------- | ---------- | -------------- |
> | LoRA r=8    | 5.08GB         | 22.26GB         | 69.73      | 34.84          |
> | DoRA r=8    | 5.37GB         | 22.87GB         | 71.46      | 35.47          |
> | LoRA r=60   | 5.28GB         | 22.98GB         | 69.78      | 34.92          |
> | DoRA r=60   | 5.58GB         | 23.61GB         | 71.78      | 35.65          |
> | SalientLoRA | 5.41GB -> 5.12GB    | 23.12GB -> 22.43GB  | 72.68      | 36.22          |
>
> ***Q2: Based on my understanding, the constructed influence domains form an undirected simple graph. If this graph forms a single cycle, how do you perform de-cycling?***
>
> In practice, the constructed dependency graph of singular values is a directed graph, where nodes represent singular values and edges represent their dependencies. This can be explained from two aspects.
>
> (1) The dependencies are quantified through slopes between two singular values within a time-series, where the choice of independent and dependent variables significantly influences the slope calculations. For instance, consider two singular values, $\lambda_a$ and $\lambda_b$. If $\lambda_a$ is chosen as the independent variable and $\lambda_b$ as the dependent variable, the resulting slope $k_{ab}$ quantifies the influence of $\lambda_a$ on $\lambda_b$. Conversely, selecting $\lambda_b$ as the independent variable results in the slope $k_{ba}$. Since $k_{ab}$ and $k_{ba}$ are numerically distinct, the weights of the edges from $\lambda_a$ to $\lambda_b$ and $\lambda_b$ to $\lambda_a$ also differ, thus forming a directed graph.
>
> (2) Intuitively, the dependency relationship between two singular values is unidirectional: if a singular value is of significant importance, the variations of its magnitude can impact changes in the other one, reverse is not necessarily true.
>
>
> ***Q3：Do you calculate influence domains starting from vertices with a degree of 0, similar to topological sorting, and then update the degrees of the vertices connected to it, then repeat the process***
>
> The calculation of influence domains resembles a reverse topological sort. We calculate the influence domains of all vertices by proceeding backwards through the graph. Specifically, the process initiates from vertices with an out-degree of zero. These vertices do not influence any other nodes, so their influence domain is directly assigned a value of 1. Subsequently, the influence domain of a node that has subsequent nodes is calculated as the weighted sum of the influence domains of these downstream nodes. This iterative procedure continues until the influence domains for all nodes in the graph have been determined.

---

> > ### Author Response · Authors · 2024-08-13
> >
> > Dear Reviewer xsy4,
> >
> > Thank you for your thorough and valuable feedback. We have carefully addressed each of your concerns. We hope that our clarifications provide satisfactory answers to your questions. We are committed to improving our work based on your insights, and we look forward to your response.
> >
> > Best regards,
> >
> > The Authors

---

### Author Rebuttal · Authors · 2024-08-07

***1. A Detailed Explanation of the De-cycling Process.***

Since the dependency graph between singular values is a directed cyclic graph, we use a depth-first search (DFS) algorithm to detect and remove cycles. Specifically, we begin by performing a depth-first traversal of each node in the graph, recording the path in a stack. If a node is encountered that is already present in the path, a cycle is detected. At this stage, we remove the edge with the smallest weight from the cycle. This process is repeated until all nodes have been traversed.

 The pseudocode is provided in the following PDF.


***2. A Thorough Analysis of Salience Measurement***

The salience measurement consists of two components: the magnitude of singular values and the influence domain.

(1) Magnitude of Singular Values

After performing singular value decomposition (SVD), the singular values represent the primary characteristics of a matrix. Larger singular values contain more information from the matrix and capture more of its essential features, thereby possessing greater importance. Consequently, singular values play a dominant role in the SVD matrix. In light of this, our method utilizes the magnitude of these singular values to assess their significance, and combines this with the orthogonality loss at each timestep to produce a more robust evaluation.

(2) Influence Domain

Numerous studies [1, 2] demonstrate that there exists the interdependencies among the different structures of models. Constructing a dependency graph of these structures is instrumental in analyzing the importance of model structures. Inspired by this, we assume that the intrinsic ranks among different matrices also possess certain dependencies and influence relationships. The intrinsic rank of important  matrices can exert influence over other matrices. Therefore, beyond the magnitude of singular values, we analyze the intrinsic dependencies among singular values in different matrices to measure their importance. These dependencies reveal how variations in singular values influence each other across multiple timesteps. If variations in a singular value can induce changes in several others, it indicates that the singular value has a broader influence domain and is of higher importance.

To intuitively reveal the influence domain of singular values, we visualize their distributions across various datasets, as shown in the following PDF. These distributions exhibit similar characteristics, with deeper layers and FFN (Feedforward Neural Network) layers having a larger influence domain. This aligns with findings from previous studies [3, 4], which indicate that deeper layers and FFNs are more significant in the model.

------
[1] LLM-Pruner: On the Structural Pruning of Large Language Models. Xinyin Ma and Gongfan Fang and Xinchao Wang, 2023. In NeurIPS.

[2] LoRAShear: Efficient Large Language Model Structured Pruning and Knowledge Recovery. Chen et al., 2023. In arXiv preprint.

[3] LoRAPrune: Pruning Meets Low-Rank Parameter-Efficient Fine-Tuning. Zhang et al., 2024. In ACL.

[4] AdaLoRA: Adaptive Budget Allocation for Parameter-Efficient Fine-Tuning. Zhang et al., 2023. In ICLR.

---

### Author Response · Authors · 2024-08-14
**Explaination of the Innovations and Contributions of Paper ID 7458: Unveiling LoRA Intrinsic Ranks via Salience Analysis**

Dear Chairs, SACs,  ACs, and Reviewers,

We appreciate the opportunity to discuss our NeurIPS submission titled "Unveiling LoRA Intrinsic Ranks via Salience Analysis". We are grateful for the evaluations provided by the review committee. As a foundational study, we would like to highlight the key strengths and innovations of our work, as we hope that our work will not only advance the field of parameter-efficient fine-tuning (PEFT), but also offer valuable insights for future research.

**1. Novelty and Contribution:** We propose a novel intrinsic rank allocation and fine-tuning method, SalientLoRA. This method adaptively allocates ranks based on matrix importance, revealing the intrinsic rank of LoRA via salience analysis, and achieves noteworthy fine-tuning performance with high efficiency. We validate the fine-tuning effects on models with different architectures (encoder-only, encoder-decoder, decoder-only) across multiple tasks, demonstrating performance improvements of 0.96%-3.56% over state-of-the-art (SOTA) baselines. Notably, compared to other rank allocation methods like AdaLoRA [1], our method reduces fine-tuning time by up to 94.5%. This high efficiency aligns with the primary goals of parameter-efficient fine-tuning methods.

**2. Generalizability of Insight:** Our work suggests promising potential for generalizability, presenting a foundational study that could enhance both the efficiency and memory usage of models during fine-tuning. Specifically, as an adaptive rank allocation approach, our method can be combined with other LoRA-based fine-tuning methods, such as DoRA [2], which applies LoRA to different architectures. By integrating our SalientLoRA with these approaches, our method is capable of uncovering the intrinsic rank of LoRA, thereby further enhancing fine-tuning performance. Given the broad applicability, we anticipate that our work could make a meaningful contribution to the research field.

We appreciate the efforts of Chairs, SACs, ACs and reviewers in guiding this process. We hope that the innovative aspects and the potential impact of our work can be fully recognized in the final deliberations.

Thank you once again for your consideration and the opportunity to contribute to NeurIPS.

Best regards,

The authors

---

**References**

[1] AdaLoRA: Adaptive Budget Allocation for Parameter-Efficient Fine-Tuning. Zhang et al., 2023. In ICLR.

[2] DoRA: Weight-Decomposed Low-Rank Adaptation. Liu et al., 2024. In ICML.

---

### Decision · Program_Chairs · 2024-09-25

**Decision:**

Accept (poster)

**Comment:**

This paper proposes a new method, SalientLoRA, for adaptively optimizing the intrinsic rank of low-rank adaptation (LoRA) matrices. Experiments on NLU, NLG, and instruction tuning tasks show that SalientLoRA achieves state-of-the-art fine-tuning performance and balances fine-tuning efficiency and performance. The proposed SalientLoRA method is novel and the experimental results are encouraging.

This paper received borderline scores and the scores did not improve after discussion between the authors and reviewers. The reviewers' main concerns include: 1) the paper lacked a detailed analysis of hyperparameters and 2) the paper was not clearly written (e.g., lack of details, with clarity issues). The authors attempted to address these issues by providing some analysis and details in the rebuttal. The reviewers believe that by adding the responses to reviewer i7vj to Section 6.7, the analysis of the impact of various hyperparameters can be considered satisfactory. However, even with the inclusion of the details provided by the authors at the rebuttal stage (e.g., an explanation of the de-cycling method), the writing of this paper still does not reach the level that reviewers expect for a top-level paper. If this paper is accepted, the writing needs to be significantly revised and improved.